# Alpha-Emitter Radiopharmaceuticals and External Beam Radiotherapy: A Radiobiological Model for the Combined Treatment

**DOI:** 10.3390/cancers14041077

**Published:** 2022-02-21

**Authors:** Anna Sarnelli, Maria Luisa Belli, Irene Azzali, Emiliano Loi, Stefano Severi, Lidia Strigari

**Affiliations:** 1Medical Physics Unit, IRCCS Istituto Romagnolo per lo Studio dei Tumori (IRST) “Dino Amadori”, 47014 Meldola, Italy; anna.sarnelli@irst.emr.it (A.S.); emiliano.loi@irst.emr.it (E.L.); 2Biostatistics and Clinical Trials Unit, IRCCS Istituto Romagnolo per lo Studio dei Tumori (IRST) “Dino Amadori”, 47014 Meldola, Italy; irene.azzali@irst.emr.it; 3Nuclear Medicine Unit, IRCCS Istituto Romagnolo per lo Studio dei Tumori (IRST) “Dino Amadori”, 47014 Meldola, Italy; stefano.severi@irst.emr.it; 4Department of Medical Physics, IRCCS Azienda Ospedaliero-Universitaria di Bologna, 40138 Bologna, Italy; lidia.strigari@aosp.bo.it

**Keywords:** mCRPC, radium-223 chloride, α-particle therapy, combined therapies, EBRT, RBE

## Abstract

**Simple Summary:**

The linear–quadratic (LQ) model was adapted to ^223^Ra therapy using brachytherapy formalism for a mixture of radionuclides, considering the contribution of all daughter isotopes in the decay chain. The LQ model allowed us to predict the two-year overall survival and neutropenia rates with a combination of external beam radiotherapy and ^223^Ra treatment. The fitted model could be used to guide future optimization and personalization of combined treatments.

**Abstract:**

Previously published studies combined external beam radiotherapy (EBRT) treatments with different activities of ^223^Ra. The data of two-year overall survival (2y-OS) and neutropenia (TOX) incidence when combining EBRT and ^223^Ra are not homogeneous in literature. We adapted the linear–quadratic model (LQ) to ^223^Ra therapy using brachytherapy formalism for a mixture of radionuclides, considering the contribution of all daughter isotopes in the decay chain. A virtual cohort of patients undergoing ^223^Ra therapy was derived using data from the literature. The doses delivered using ^223^Ra and EBRT were converted into biologically equivalent doses. Fixed-effect logistic regression models were derived for both the 2y-OS and TOX and compared with available literature. Based on the literature search, four studies were identified to have reported the ^223^Ra injection activity levels varying from the placebo (0) to 80 kBq/kg, associated or not with EBRT. Logistic regression models revealed a dose-dependent increase in both the 2y-OS (intercept = −1.364; slope = 0.006; *p*-value ≤ 0.05) and TOX (−5.035; 0.018; ≤0.05) using the EBRT schedule of 8 Gy in 1 fr. Similar results were obtained for other schedules. Discrepancies between our TOX model and those derived for EBRT combined with chemotherapy are discussed. Radiobiological models allow us to estimate dose-dependent relationships, to predict the OS and TOX following combined ^223^Ra + EBRT treatment, which will guide future treatment optimization.

## 1. Introduction

Prostate cancer (PCa) is the most common cancer in men and the second leading cause of cancer death. In patients with advanced PCa, metastases usually occur in the lymph nodes and bones (>300,000 cases per year in the United States) [1]. The initial treatment for metastatic PCa patients is medical castration. Still, its clinical benefit is temporary, and most patients develop metastatic castration-resistant PCa (mCRPC) disease [2,3]. This specific pathology is associated with a poor outcome, making metastatic prostate cancer a prime target for bone-targeted therapies using calcium analogs.

In 2013, Radium-223 chloride (^223^RaCl_2_) was approved by the FDA for bone metastases treatment of mCRPC patients. ^223^RaCl_2_ is a first-in-class radionuclide pharmaceutical (RN) α-emitter that has intrinsic bone-targeting properties. The radium mimics the behavior of calcium distribution. It accumulates in proliferating cells in and around bone metastases, with a high affinity for the bone matrix [4]. The short range of α-particles in tissue means local, highly precise ^223^RaCl_2_ accumulation in the target lesion is possible, with a relative biological effectiveness (RBE) of about five compared to gamma-irradiation [5]. The combination of its ultrashort range, high intensity, highly efficient bone localization, and RBE make ^223^RaCl_2_ an excellent candidate for targeted molecular radiation therapy.

According to these features, a series of phase one, two, and three clinical trials explored the therapeutic role of ^223^RaCl_2_ for bone metastases. Among these clinical studies, the ALSYMPCA trial, an international placebo-controlled phase-three study, enrolled 921 mCRPC patients with symptomatic bone disease. The ALSYMPCA results and those of other recently published studies [6,7,8,9,10] demonstrated that ^223^RaCl_2_ use improves the overall survival (OS) and delays the time to the first symptomatic skeletal event versus the placebo.

^223^RaCl_2_ was also combined to external beam radiotherapy (EBRT) to enhance treatment efficacy by delivering a higher target radiation dose.

Worldwide, clinical studies have empirically proven the efficacy of this treatment approach. Nevertheless, the optimal activity and treatment schedule for combined (RN + EBRT) treatments have not been definitively ascertained.

There are several limits to our progress with determining the potential advantages of α-based RN treatments and their combination with EBRT, including the lack of effort to personalize these treatments (e.g., injection activity based on a patient’s weight rather than tailored based on predictive dosimetry) and the scarcity of dosimetric data associated with the outcome of the combination RN. At the same time, numerous α-emitter RNs are under development with the intention of those becoming commercially available on the market to be used in the coming years.

In addition, the linear–quadratic (LQ) model was recently adapted to ^225^Ac-PSMA (prostate-specific membrane antigen) therapy using brachytherapy formalism for a mixture of radionuclides, considering the contribution of all daughter isotopes in the decay chain [11]. This model could also be applied to ^223^RaCl_2_ therapy to improve the accuracy of biologically effective dose (BED) calculation. Thus, to combine low-LET (linear energy transfer) EBRT and high-LET RN absorbed doses, we adapted the radiobiological model recently published for α-particle emitters to determine radiobiological quantities, such as those of BED, and for 2 Gy equieffective dose (EQD2) calculation [11,12].

Moreover, in the era of the digital twin and treatment personalization, we attempted to develop a virtual cohort of patients treated with ^223^RaCl_2_ and EBRT, to determine the above radiobiological quantities associated with the OS and toxicity rates.

Based on the above assumptions, this work aimed to assess the dose-dependent relationships predicting the OS and toxicity following the combination of RN and EBRT treatments.

## 2. Materials and Methods

### 2.1. Study Data Search

Relevant literature was reviewed to extract information on treatment outcomes in terms of both efficacy and safety from randomized phase 2–3 studies. Regarding treatment efficacy, the two-year OS rate (2y-OS) was assumed as the endpoint parameter to assess the efficacy of combined therapy. The data for the 2y-OS were extracted either from Kaplan–Meier plots or, if they were not available, from the information reported in the main text.

Regarding treatment safety, cumulative bone marrow damage is well-recognized as representing the toxicity of radiopharmaceuticals. Treatment with ^223^Ra was associated with a greater risk of neutropenia and thrombocytopenia. To model and predict the toxicity (TOX) endpoint, we focused on the neutropenia incidence. The information was extracted from data reported in each considered study.

To identify relevant studies, we searched two databases (Medline and Scopus). The inclusion criteria for study selection were as follows: (1) all patients should be affected by mCRPC; (2) combined treatment of targeted ^223^Ra α-therapy and EBRT; (3) randomized clinical trials; (4) published in the last ten years; (5) 2y-OS and/or neutropenia rate reported.

Reviews were also screened for additional papers. All data were checked for internal consistency and compared with data published in related papers. The complete search strategy for every database is reported in Appendix A. For every search record, three independent researchers screened the title and abstract for potentially evaluable studies.

### 2.2. Radiobiological Model

Since the dose-limiting factor in toxicity is the red marrow irradiation, as most radio-sensible normal tissues are positioned close to bone lesions, the absorbed dose of bone metastasis for both therapies (RN and EBRT) was used for both efficacy and safety evaluation.

#### 2.2.1. Radiobiological Model for Low-LET EBRT

The LQ model was used to evaluate the BED for EBRT [12,13]:(1)$$BED_{EBRT}=D\left[1+\frac{D/n_{f}}{\alpha /\beta }\right]\;$$
where *D* is the prescribed dose, *n_f_* is the number of fractions, and *α* and *β* are the radiobiological parameters of the LQ model. Then, *BED_EBRT_* was converted into *EQD2_EBRT_* as follows [12,13]:(2)$$EQD2_{EBRT}=BED_{EBRT}\left(1+\frac{d}{\alpha /\beta }\right)$$
where *d* = 2 Gy.

#### 2.2.2. Generation of Virtual Cohort of Patients Treated with ^223^RaCl_2_

Since no dosimetric evaluations were reported in the selected studies, we derived the absorbed dose estimation from Pacilio et al. [14], who performed a dosimetric evaluation of 24 lesions (from 9 patients) based on planar images using the MIRD formalism [15,16,17]. Additional details of the dosimetric methodology we used are reported in [14].

For each patient, the absorbed dose for each lesion mass reported in Pacilio et al. [14] was fitted with exponential curve fitting. We used the spherical model for tumor dose evaluation.

In addition, we recalculated the residence time τ for each lesion as described in Appendix B. In our study, we assumed the equilibrium between ^223^Ra and all the daughters present in the decay chain. As such, the residence time τ calculated for ^223^Ra was assumed valid for all daughters in the decay chain. Then, we calculated the non-weighted RBE absorbed dose for each daughter in the decay chain (Figure 1) using the spherical-like tumor model implemented in OLINDA/EXM software (version 1.1, Vanderbilt University, Nashville, TN, USA [18]).

#### 2.2.3. Radiobiological Model for High-LET α-Particle RN ^223^RaCl_2_

As proposed in a previously published paper [11], the following formalism was used to account for the mixture of radionuclides present in the ^223^Ra decay chain (Figure 1) with a continuous and exponentially decreasing dose-rate of high-LET α-emitters during therapy:(3)$$R_{i}\left[\frac{mGy}{h}\right]=D_{i}\left[\frac{mGy}{MBq}\right]*A_{inj}\left[MBq\right]*\lambda \left[\frac{1}{h}\right]$$
(4)$$BED_{RN}=n_{c}\left(\frac{1}{\lambda }\underset{i}{\sum }R_{i}\right)*\left\{RBE_{max}+\frac{\sum_{i}\sum_{j}R_{i}R_{j}}{\left(\alpha /\beta \right)*\left(\mu +\lambda \right)*\left(\sum_{i}R_{i}\right)}\right\}$$
where *i* and *j* stand for each radionuclide present in the decay chain (i.e., *i*, *j* = ^223^Ra, ^219^Rn, ^215^Po, ^211^Pb, ^211^Bi, ^211^Po, ^207^Tl), *R_i_* is the dose rate, *D_i_* is the non-weighted RBE mean absorbed dose for injection activity calculated with the spherical model of OLINDA/EXM software, *A_inj_* is the injected activity of ^223^RaCl_2_, λ is the effective half-life of ^223^Ra (assumed equal for all daughters in the decay chain), *μ* is the constant of sublethal damage repair, and *n_c_* is the number of cycles. As the branching ratio of beta ^215^Po decay is <0.001%, this decay was omitted from the calculation and a full alpha decay in ^211^Pb was assumed for dose estimation.

For the high-LET α-particle, we used the formalism proposed by [11,19] to calculate the maximum value of RBE (*RBE_max_*) as follows:(5)$$RBE_{max}=RBE_{exp}+\frac{d}{\alpha /\beta }\left(\frac{RBE_{exp}^{2}-1}{RBE_{exp}}\right)$$
where *RBE_exp_* is the experimental value of RBE for α-particles reported in the literature and *d* is the dose per fraction of a reference low-LET therapy.

Table 1 summarizes the radiobiological parameters considered for the *BED_RN_* calculation.

Based on Equation (3) and the parameters reported in Table 1, the RBE_max_ value was equal to 5.96. The *BED_RN_* value was calculated for each patient subgroup based on the *A_inj_* reported in the selected studies. Then, the BED_RN_ was converted to an *EQD2_RN_* as follows [12,13]:(6)$$EQD2_{RN}=BED_{RN}\left(1+\frac{d}{\alpha /\beta }\right)$$

The mean *EQD2_RN_* value obtained for the 24 considered lesions was calculated as representative of the entire group for each subgroup of administered ^223^RaCl_2_ injections.

### 2.3. Radiobiological Model of Combined Therapy

The *EQD2_TOT_* from combined treatments of RN and EBRT was determined for all the considered patient cohorts. For each subgroup with a defined ^223^RaCl_2_ activity, the *EQD2_TOT_* was obtained by considering the proportion of dose received from EBRT and RN therapy, as follows:(7)$$EQD2_{TOT}=EQD2_{RN}+f*EQD2_{EBRT}$$
where *f* is the percentage of patients receiving EBRT.

### 2.4. Treatment Outcome Modeling and Model Comparison

Mixed-effect logistic regression was used to model the effect of the dose on the outcome after applying sample weights according to the sample size. Models other than a simple regression allow us to give more influence to larger studies and to account for residual heterogeneity among outcomes not modeled by the dose.

For all analyses, *p*-values of <0.05 were considered statistically significant.

All analyses were performed using R software (version 4.0.4).

Then, the derived model of the toxicity impact of neutropenia for the combined treatment (^223^Ra + EBRT) was graphically compared to previously published models for the outcome of EBRT treatment in combination with chemotherapy [20,21,22].

## 3. Results

The initial search yielded 25 unique citations. Four studies met the inclusion criteria and were included in this analysis (Table 2). More than 1000 patients were enrolled in the identified phase two and three clinical studies.

Regarding RN therapy, different injection activity levels were used: placebo (0), 25, 50, 55, and 80 kBq/kg. *BED_RN_* calculation was performed for each group (Table 2).

Though different EBRT schedules have been reported in the literature, the number of patients who underwent each schedule was not specified [9]. We considered three different scenarios, each with varying schedules of EBRT. The EBRT schedules were: 30 Gy delivered in 10 fractions (3 Gy/fr); 20 Gy delivered in 5 fractions (4 Gy/fr), and; 8 Gy delivered in 1 fraction (8 Gy/fr), corresponding to BED_EBRT_ (EQD_2EBRT_) values of 39.0 Gy (32.5 Gy), 28.0 Gy (23.3 Gy), and 14.4 Gy (12.0 Gy), respectively. The fraction *f* as a percentage of patients receiving EBRT is indicated in Table 2 for each study.

Table 3, Table 4 and Table 5 summarize the EQD2 calculated for each group with the above-reported EBRT schedules, assuming an α/β ratio of 10 Gy for both tumors and acute effects.

The proportion of surviving patients up to two years (2y-OS) was extracted from each study. It should be noted that the 2y-OS information was extracted with different modalities. Studies A and B reported the number of events or patient survival; thus, the 2y-OS was estimated assuming no censored patients. On the other hand, in studies C and D, the 2y-OS was extracted directly from Kaplan–Meier curves. Therefore, for studies C and D, censored data were included in the provided information.

We investigated the EBRT schedules’ impact on the outcomes. For the sake of simplicity, we report the results for the EBRT schedule of 8 Gy in 1 fraction (fr). The results for the other two considered EBRT schedules (i.e., 20 Gy in 5 fr and 30 Gy in 10 fr) are reported in Appendix A.

The logistic regression models revealed no significant effect of the EBRT schedule on the TOX (*p* = 0.05), while the EBRT schedule significantly impacted the 2y-OS (*p* = 0.006). Comparison models are shown in Figure 2, while the model parameters are summarized in Table 6.

A graphical comparison of our logistic regression model of hematological toxicity with the previously published models is shown in Figure 3. Four different models were considered for the comparison [16,17,18], where hematological toxicity of grade three or above was described in terms of the mean EQD2 dose absorbed by the red marrow, delivered by EBRT combined with chemotherapy. One study [18] derived different models for two different chemotherapy regimens (5FU and FOLFOX) associated with the same EBRT. The graphical comparison clearly shows the lower toxicity profile of combined ^223^Ra + EBRT treatment compared to models derived for EBRT in combination with chemotherapy [20,21].

## 4. Discussion

The combination of a high-LET (60–300 keV/µm) and short-range (40–100 µm in tissues) α-particle [23] with a low-LET EBRT approach allowed us to increase the survival rates when treating a disease that resists other therapies [24].

The results for ^223^RaCl_2_ were promising. The ALSYMPCA study demonstrated a more prolonged median survival of three months for patients treated with ^223^Ra than the placebo, with a reduced time to symptomatic skeletal events [7]. While α-therapy can have a highly destructive effect at the systemic level, EBRT treatment can provide a booster dose localized to a specific region.

The data reported in literature indicates that it is possible to increase tumoral cell killing and survival with combined α and photon treatments [24].

Regarding the toxicity profile, neutropenia is a common complication of both EBRT and ^223^Ra therapy. In this context, several authors reported that EBRT during or after ^223^RaCl_2_ remained significantly associated with an increased risk of bone marrow failure [25], while other authors reported that “EBRT did not adversely affect ^223^Ra hematologic safety” in sub-group analysis of the ALSYMPCA trial [26].

Thus far, we have not yet developed a radiobiological model describing the dose-dependent behavior of the combined RN and EBRT option using the available clinical data.

Biophysical patient-specific models allow us to encode the known physiological behavior within mathematical equations and tune these models to represent individual patients. Our study aimed to create these digital twins and apply the virtual patient cohort to model and predict the OS and TOX incidence in published studies. While conceptually simple, the reality of determining the equations in practice, tuning the parameters to patient data, and generating reliable predictions remains significantly challenging due to the scarcity of reported patient information in the literature. Yet, there is huge potential if we can overcome these challenges. Once a patient-specific model is created, it can then be re-used to design and personalize new RN treatments.

This study investigated a pooled virtual cohort of patients undergoing ^223^Ra to derive the dose–effect correlations of the combined α and EBRT dose delivery. A radiobiological model was proposed to describe this dependence and predict the 2y-OS and TOX incidence.

The logistic regression from combined RN and EBRT treatment approaches, calculated using several hypotheses of EBRT schedules, suggests a dose-dependent increase in 2y-OS incidence (β = 0.006, *p*-value = 0.006, Figure 2 using the model parameters of Table 6) with a slight effect of the total dosage.

As recently reported, the dose–effect relationship for tumor-control probability suggests a minimum cut-off of 20 Gy for the absorbed dose to guarantee stable disease with a partial or complete response [27]. Therefore, in the case of suboptimal dose delivery using the RN, EBRT is an evidence-based and highly cost-effective strategy to increase the local dose to target areas and consequently improve the OS.

The association of neutropenia with the mean absorbed dose delivered via combined EBRT + RN treatment was confirmed in our study, highlighting that the neutropenia incidence slightly increases with the given absorbed dose.

Our data show that the combined therapy does not strongly increase the toxicity profile, and it has an advantage in terms of the 2y-OS for the overall population. However, considering the limited data available and the impossibility of differentiating the EBRT and RN information at the patient level, only our methodological comparison with the previously published NTCP model for hematological toxicity was applicable. Detailed correlations between clinical data (in terms of toxicity and survival fraction) with absorbed dose estimation stratified over homogeneously treated patient populations would provide helpful information for dose escalation of ^223^Ra treatment.

We showed that increasing the mean absorbed dose to the target will increase the 2y-OS by up to about 50%. The same increase in the mean absorbed dose is associated with an acceptable toxicity profile, which only increases up to about 20%. We also showed that the dose increase is mainly guided by ^223^Ra treatment, as EQD2_TOT_ does not strongly depend on EBRT when considering a schedule providing EQD2_EBRT_ up to 39 Gy.

Advancements in radiotherapy have allowed for the delivery of high-precision, dose-escalated treatment, known as stereotactic body radiotherapy (SBRT), to target lesions throughout the body with excellent rates of local control in oligometastatic patients [28]. Delivering high-dose, single-fraction SBRT seems to be an effective treatment option for patients with painful bone metastases, associated with higher rates of pain response in evaluable patients following conventional schedules [29]. Thus, the combination of α-therapies with the sharp dose falloff of SBRT approaches and multidisciplinary consultation represent new opportunities for appropriately selected patients. In other words, to tune the combined RN and EBRT therapies in terms of treatment outcome, in principle, we must balance the ^223^Ra treatment in terms of the absorbed dose and perform a dose-escalation using SBRT.

Nowadays, personalized dosimetry is feasible [14,27,30,31] although not routinely performed. A dosimetric approach to ^223^Ra may help us to better understand the impact of combining SBRT and novel α-based RN treatments [32].

Regarding the TOX models’ comparison, we showed a discrepancy between our model and the previously published models derived from (chemo)EBRT treatments. Higher grades of toxicity (i.e., G3 or more) are generally expected at higher doses than those normally related to G2 or more. These discrepancies might be related to the large irradiated volumes reported in the works of Bazan et al. [20] and Yoshimura et al. [21] for the anal canal and gynecological malignancies, respectively. In other words, for the TOX incidence, the dose and volume should be considered to optimize the plan. These discrepancies could be also explained by the different microscopical patterns of local damage delivered with α-particles compared to photon/electron irradiation.

Model comparison also suggests a possible role of chemotherapy in increasing toxicity. This observation agrees with higher toxicity in patients who have undergone previous cytotoxic treatments [27]. Moreover, Sciuto et al. [27] recently showed a correlation between the mean absorbed dose of ^223^Ra and chromosome damage in the number of dicentrics and micronuclei, which are related to the risks of secondary malignancies. This aspect should also be considered for diagnostic administration, as discussed in [33], to better assess/prevent the risk of secondary effects.

In our study, the ^223^Ra biokinetic was assumed valid for all daughters in the decay chain. This assumption may not be valid, as the daughters present in the decay chain may differ on the father’s retention and extraction pathways or bone areas [30]. Neglecting daughters’ biokinetics may overestimate the absorbed dose by about 18% [30].

Moreover, from the radiobiological point of view, we assumed that the repair constant (*μ*) was independent of the level of damage. At the same time, it has been argued that it may decrease with an increasing dose or dose rate [19].

Regarding RBE, the estimated value of *RBE_max_* = 5.96 agrees with the theoretical value equal to 5.6 [5] calculated for ^223^Ra by a weighted sum of RBEs for the individual emissions, and with the experimental value of 5.4 [34]. At present, there is no consensus on the RBE values to adopt for α-particles. Published studies report different values of RBE for α-particles, but this discrepancy may arise from the different endpoints considered in each study for RBE evaluation. Historically, in the absence of in vivo clinical data, RBE = 5 was assumed precautionary based on the deterministic effect evaluated via in vitro experiments on cell-surviving fractions (37% surviving fraction) that produced an RBE value ranging from three to five [35,36]. Yet, the details of the in vivo measurements based on which RBE = 5 was suggested as a conservative assumption are not reported in the literature [35]. Other studies reported by Sgouros et al. [5] considered the 37% surviving fraction of MDA-MB-231 cells irradiated with ^213^Bi and produced RBE values ranging from 3.7 to 15.6, with a median value of 4.7. On the other hand, Howell et al. [34] considered the in vivo measurement of survival of murine testicular sperm heads irradiated with ^223^Ra, with α-particles’ energies ranging from 3 to 9 MeV, and found RBE = 5.6. In vivo estimation of the RBE value for α-particles may provide further detail for the radiobiological model of α-particle RN therapy.

Different assumptions were adopted in our study, which should be considered when evaluating the obtained results. As specified above, the data for the 2y-OS were produced via different modalities in each study, based on the available information. For study A, the information on surviving patients at two years was reported in the main text as the proportion of the whole population. Therefore, for study A we consider the data on the 2y-OS to be accurate. Then, the text documented the proportion of patients who died at two years for study B. Assuming no data were missing, we then calculated the rate of patient survival for the whole patient population. In doing so, we may have overestimated the 2y-OS data. For studies C and D, the data were manually extracted from Kaplan–Meier curves, which means there is a degree of uncertainty surrounding the accuracy of the estimates.

In addition, patient-specific data (e.g., EBRT schedules, clinical information, etc.) from each considered study were missing. In our investigation, we assumed different EBRT schedules to account for different treatment schedules uniformly across the population. However, this assumption may not be valid as patients who received EBRT treatment with different EBRT dose prescription regimens were generally classified as patients receiving combined ^223^Ra and EBRT treatment in each study. The weighted averaged dose was used to describe the absorbed dose associated with each patient, to account for combined treatments over a population with mixed regimens.

Regarding RN cycles, in study D [37], the number of cycles was not equal among the patients. The authors reported that “up to 6 cycles were delivered”. We assume six cycles were delivered to all considered patients. While for the toxicity, this assumption impacts the data conservatively, this may affect treatment efficacy and thus impact survival. Moreover, a portion of patients received EBRT after RN treatment, and others before. The time of delivery may be an additional confounding factor. Therefore, detailed information on EBRT schedules and treatment outcomes at the patient level is expected to further improve the accuracy and precision of the proposed model.

Furthermore, we assumed a single-phase biokinetic for the ^223^Ra bone uptake, as in the work of Pacilio et al. [14]. However, a two-phase bone-seeking biokinetic was recently reported in the literature by Taprogge et al. [38] based on six patients and two administrations of ^223^RaCl_2_. The two-phase bone model showed a fast first uptake phase at the bone surface, with an uptake phase from the plasma to bone phase one: 4.0 h^−1^ (range: 1.9–10.9) and a washout phase from bone phase one to the plasma: 0.15 h^−1^ (range: 0.07–0.39). Then, activity slowly migrated into the second compartment, which had a prolonged release rate with an uptake phase from bone phase one to bone phase two of 0.03 h^−1^ (0.02–0.06) and a washout phase from bone phase two to bone phase one: 0.008 h^−1^ (0.003–0.011). The authors found a maximum skeletal uptake of 49% at four hours post-injection, besides the fast first uptake phase, in agreement with Yoshida et al. [39]. The impact of the two-phase model was not considered in our study. This could potentially result in ^223^Ra localization in different bone areas, rather than a homogenous distribution on the bone surface, as assumed in our study.

## 5. Conclusions

Our radiobiological model describes dose-dependent relationships to predict the OS and TOX following combined treatment regimens with ^223^Ra and EBRT. Information previously described in the literature was used to construct a virtual patient cohort and extract the parameters necessary to build a mathematical model that was useful for whole-treatment description. This development could help both to support a global description of patient outcomes and guide future treatment optimization in combined regimens with RN and EBRT.

## Figures and Tables

**Figure 1 cancers-14-01077-f001:**
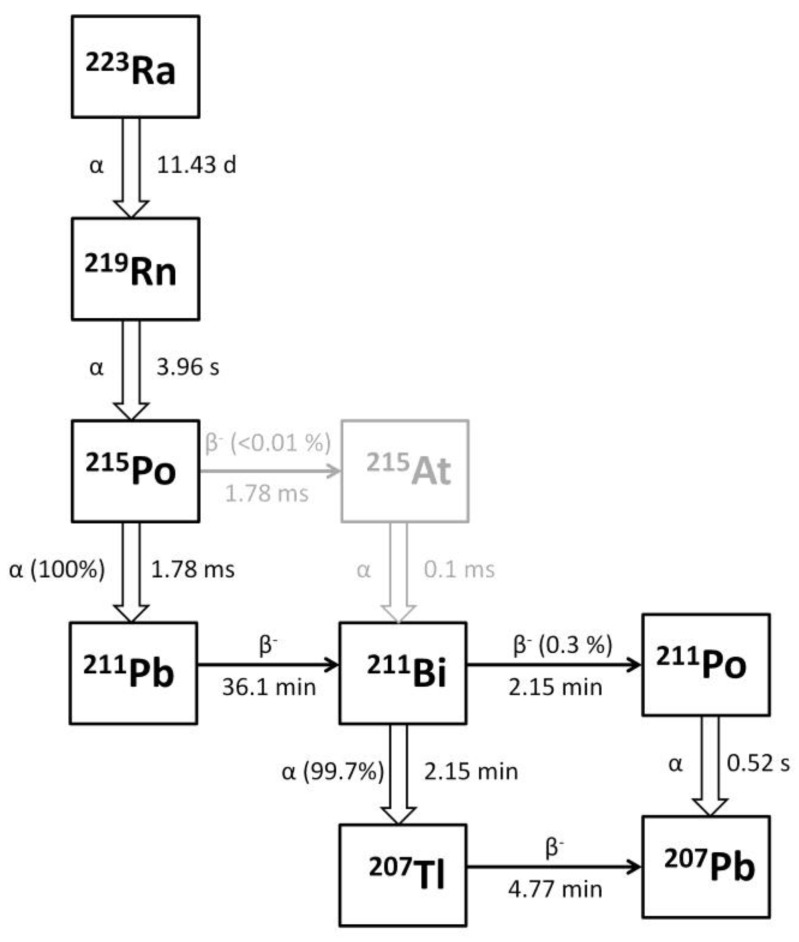
Schematic representation of ^223^Ra decay chain.

**Figure 2 cancers-14-01077-f002:**
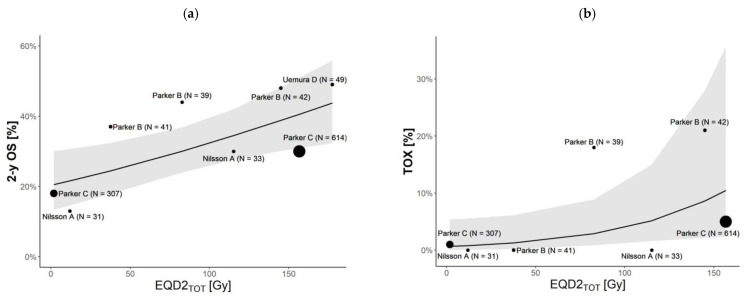
Logistic regression model for (**a**) two-year overall survival (2y-OS) and (**b**) toxicity impact (TOX) measured as neutropenia rate. Predicted values are reported at the study level (fixed-effect only). The dots’ dimensions are proportional to the number of patients included in the study (Table 2). The grey area represents the 95% CI. Models’ parameters are reported in Table 6. Note that study A (Nilsson et al. [8]) with a sample of 31 and study C (Parker et al. [7]) with a sample of 307 are both ^223^Ra placebo studies. For the sake of simplicity, we report the results of the EBRT schedule of 8 Gy in 1 fr. The results for the other two EBRT schedules we considered (i.e., 20 Gy in 5 fr and 30 Gy in 10 fr) are reported in Appendix A.

**Figure 3 cancers-14-01077-f003:**
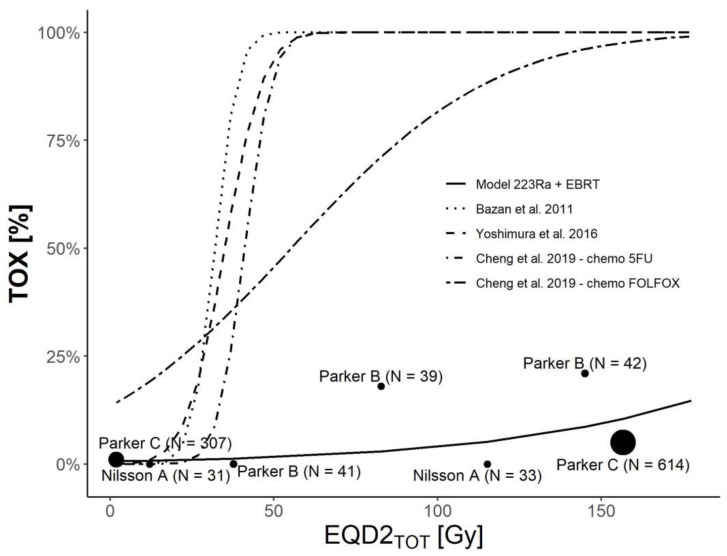
Our logistic regression model of the toxicity impact (measured as the neutropenia rate (TOX) for combined therapies (^223^Ra + EBRT)) compared to a previously published model of EBRT in combination with chemotherapy. The dots indicate the data for each considered study (Table 2). The dots’ dimensions are proportional to the number of patients included in the study. Model parameters are reported in Table 6. The study of Cheng et al. [22] considered two different chemotherapy regimens (5FU and FOLFOX) in combination with EBRT and derived two different models.

**Table 1 cancers-14-01077-t001:** Radiobiological parameters used for BED and EQD2 calculations.

Radiobiological Parameter	Value	Reference
α/β	10 Gy	[19]
μ	1.4 h^−1^	[19]
$RBE_{\alpha }$ experimental	5	[5]
$RBE_{\gamma }$ $\;\mathrm{and}\;RBE_{\beta }$	1	
*d*	2 Gy/fr	

Abbreviations: fr = fraction.

**Table 2 cancers-14-01077-t002:** The selected studies, number of enrolled patients (#Pts), ^223^Ra administered per cycle, number of cycles, proportion of patients receiving external beam radiotherapy (EBRT), two-year overall survival (2y-OS), and toxicity incidence (TOX) for grade-two-or-more neutropenia.

Study	Reference	Total #Pts	Phase Study	A (kBq/kg)	*n_c_*	A (MBq/Cycle)	#Pts	*f* % (#Pts) with EBRT	2y-OS (%)	TOX (%)
A	[8]	64	2	50	4	3.5	33	100% (33)	30% (10)	0% (0)
				(placebo)	0	0	31	100% (31)	13% (4)	0% (0)
B	[7]	122	2	25	3	1.75	41	29% (12)	37% (15)	0% (0)
				50	3	3.5	39	44% (17)	44% (17)	18% (7)
				80	3	5.6	42	36% (15)	48% (20)	21% (9)
C	[6]	921	3	50	6	3.5	614	16% (98)	30% (184)	5% (31)
				(placebo)	0	0	307	16% (49)	18% (55)	1% (3)
D	[10]	49	2	55	6	4.125	49	29% (14)	49% (24)	NA

Abbreviations: *n_c_* = number of cycles; *f* = percentage of patients undergoing external beam radiotherapy (EBRT); NA = data not available.

**Table 3 cancers-14-01077-t003:** EBRT schedule: 8 Gy in 1 fraction. *BED_EBRT_* = 14.4 Gy; *EQD2_EBRT_* = 12.0 Gy.

Study	Reference	Total #Pts	A (kBq/kg)	#Pts	*f* % (#Pts) with EBRT	*EQD2_RN_* (Gy)	*EQD2_EBRT_* (Gy)	*EQD2_TOT_* (Gy)
A	[8]	64	50	33	100% (33)	103.2	12.0	115.2
			(placebo)	31	100% (31)	0.0	12.0	12.0
B	[7]	122	25	41	29% (12)	34.2	3.5	37.7
			50	39	44% (17)	77.4	5.3	82.7
			80	42	36% (15)	140.7	4.3	145.0
C	[6]	921	50	614	16% (98)	154.8	1.9	156.7
			(placebo)	307	16% (49)	0.0	1.9	1.9
D	[10]	49	55	49	29% (14)	174.0	3.5	177.5

Abbreviations: *f* = percentage of patients undergoing external beam radiotherapy (EBRT); #Pts = number of patients in each group.

**Table 4 cancers-14-01077-t004:** EBRT schedule: 20 Gy in 5 fractions. *BED_EBRT_* = 28.0 Gy; *EQD2_EBRT_* = 23.3 Gy.

Study	Reference	Total #Pts	A (kBq/kg)	#Pts	*f* % (#Pts) with EBRT	*EQD2_RN_* (Gy)	*EQD2_EBRT_* (Gy)	*EQD2_TOT_* (Gy)
A	[8]	64	50	33	100% (33)	103.2	23.3	126.5
			(placebo)	31	100% (31)	0.0	23.3	23.3
B	[7]	122	25	41	29% (12)	34.2	6.8	41.0
			50	39	44% (17)	77.4	10.3	87.7
			80	42	36% (15)	140.7	8.4	149.1
C	[6]	921	50	614	16% (98)	154.8	3.7	158.5
			(placebo)	307	16% (49)	0.0	3.7	3.7
D	[10]	49	55	49	29% (14)	174.0	6.8	180.8

Abbreviations: *f* = percentage of patients undergoing external beam radiotherapy (EBRT); #Pts = number of patients in each group.

**Table 5 cancers-14-01077-t005:** EBRT schedule: 30 Gy in 10 fractions. *BED_EBRT_* = 39.0 Gy; *EQD2_EBRT_* = 32.5 Gy.

Study	Reference	Total #Pts	A (kBq/kg)	#Pts	*f* % (#Pts) with EBRT	*EQD2_RN_* (Gy)	*EQD2_EBRT_* (Gy)	*EQD2_TOT_* (Gy)
A	[8]	64	50	33	100% (33)	103.2	32.5	135.7
			(placebo)	31	100% (31)	0.0	32.5	32.5
B	[7]	122	25	41	29% (12)	34.2	9.4	43.6
			50	39	44% (17)	77.4	14.3	91.7
			80	42	36% (15)	140.7	11.7	152.4
C	[6]	921	50	614	16% (98)	154.8	5.2	160.0
			(placebo)	307	16% (49)	0.0	5.2	5.2
D	[10]	49	55	49	29% (14)	174.0	9.4	183.4

Abbreviations: *f* = percentage of patients undergoing external beam radiotherapy (EBRT); #Pts = number of patients in each group.

**Table 6 cancers-14-01077-t006:** Models’ parameters of logistic regression for both 2y-OS and TOX impact measured as neutropenia rate. For the sake of simplicity, we report the results of the EBRT schedule of 8 Gy in 1 fraction. The results for the other two EBRT schedules we considered (i.e., 20 Gy in 5 fr and 30 Gy in 10 fr) are reported in Appendix A.

	**2y-OS Model**			
	**β**	**SE**	**z Statistic**	***p*-Value**
Intercept	−1.364	0.262	−5.209	<0.001
*EQD2_TOT_*	0.006	0.002	2.737	0.006
	**TOX Model**			
	**β**	**SE**	**z Statistic**	***p*-Value**
Intercept	−5.035	1.100	−4.578	<0.001
*EQD2_TOT_*	0.018	0.009	1.962	0.050

*EQD2_TOT_* was obtained considering the ^223^RaCl2 administration of reported schedules + EBRT (8 Gy in 1 fr). β indicates the model parameters of the intercept and slope. SE = standard error.

## Data Availability

The data presented in this study are available in this article (and Appendix A).

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
