# Peer review of "Alpha-Emitter Radiopharmaceuticals and External Beam Radiotherapy: A Radiobiological Model for the Combined Treatment"

_cancers, 2022, doi:10.3390/cancers14041077_

Round 1

Reviewer 1 Report

Strigari et al. in this excellent report present their findings on the building and evaluate a model based on the combination of external beam radiotherapy with alpha-emitters therapy. While similar experimental studies have been previously reported, there has been a wide variability in understanding the overall gain. The authors use a linear-quadratic model to assess these platforms and interestingly find there to be a dose-dependent relationship for the combination therapy. The authors have done a good job discussing the limitation of their model and do not extrapolate their data (for example instances where the time of delivering these treatments is considered). Their experiential results conveyed the hypothesis in detail. All the data looks promising and the document is of a high scientific level. However, my only concern is that it lacks a lot towards the latter end of the document where the writing can be significantly improved in terms of the flow of the document and also not having a strong concluding paragraph. I have no issues with recommending its publication in Cancers once the writing aspect of the manuscript is addressed.

Author Response

Dear Reviewer, we thanks for the kind comments. We improved both the Introduction and the Discussion section, and we added a final section of Conclusions. The modifications are highlighted along the text. We hope that these modifications may comply to your requirements and improved the manuscript quality.

An extensive English revision was also performed.

Reviewer 2 Report

In their manuscript intitled "A radiobiological modelization of combined treatments advantage : Alpha-emitter radiopharmaceuticals and external beam radiotherapy", authors presented an interesting study which adapted the LQ model to 223Ra therapy for considering the contribution of all daughter isotope in the decay chain. The manuscript is interesting and well written.
I don't have any major comments to make because your study seems to me really complete. I suggest that you separate the discussion and conclusion sections to comply with the journal's guidelines and to gain clarity for the reader. Some references do not seem complete in your bibliography. 
To what extent do you think your model could be extended to other types of emitters ? 
Congratulation for this beautiful work

Author Response

Dear Reviewer, we tank for the good comments.

As suggested, we separated the Conclusion section form the Discussion one.

We also updated the reference list.

As reported in the original study were the radiobiological model was firstly proposed (Belli and Sarnelli et al, “Targeted Alpha Therapy in mCRPC (Metastatic Castration-Resistant Prostate Cancer) Patients: Predictive Dosimetry and Toxicity Modeling of 225Ac-PSMA (Prostate-Specific Membrane Antigen). Front Oncol. 2020;10:531660.”), the proposed model could be extended to all other types of radionuclides when the two main following assumption are valid: (1) equilibrium for all daughters in the decay chain; (2) no translocation during the decay between succeeding disintegrations. These assumption allow to assume the same residency time of father for all the daughter in the decay chain.

An extensive English revision was also performed.

Reviewer 3 Report

The manuscript is well organized, clear and free from any evident fundamental and metodological error.

The manuscript is easy-to-read even if the text lacks of substantial revision made by a native English: a deep revision of the synthax is imperative before publication. To guide authors, I have included a not exhaustive list of sentences which require revision.

I have identified a list of points that require revision before publication.

Abstract:  

  • treatments with different 223Ra injection activity levels à The word “different” is connected to “activity levels” and this is not syntactically correct: please revise
  • do not abbreviate fraction in the abstract

Page 2:

  • In 2013,
  • about five, compared
  • Relevant literature has been screened to obtain treatment outcomes in terms: please revise this sentence (“screened” and “obtain treatment outcomes” can be significantly improved)

Page 3:

  • Please specify what MIRD formalism is (reference?)
  • “was fitted with an exponential curve fitting of the spherical-like tumor model.” Please revise, without using tautologies.
  • OLINDA/EXM software (version 1.1) (please add a reference).

Page 4:

  • 1: an approximation is made at the level of 215-Po: the sum of beta- and alfa decays is >100%. Please revise.
  • Please consider “as follows” instead of “as following” here and along the manuscript
  • Please specify what you refer to with “above parameters”

Page 5:

  • Please improve the sentence “explore the effect of dose on outcomes”

Page 6:

  • Be aware that in studies.. please improve the sentence
  • “without knowing if censoring is present”. please improve the sentence

Page 8:

  • “The hypothesis indicates that it is possible” if I have correctly interpreted the meaning, this is not an hypothesis, rather a result.

Page 9:

  • Missing verb in “Recently reported, dose-effect relationship for tumor control probability suggesting a minimum cut-off of 20 Gy for the absorbed dose to guarantee a stable disease, partial or complete response (23).”
  • “The association of neutropenia to combined EBRT+RN treatment with the delivered dose”, please specify what you refer to with “delivered doses”; as it is, it is ambiguous (with the doses delivered as shown in tab… etc)
  • Data+ plural
  • What do you mean with “limited data available for pour analysis ” ?
  • “We showed that increasing the mean absorbed dose to the target will increase 2y-OS” please revise
  • “combined RN 310 and EBRT therapy” -- > therapies

Page 10:

  • “be considered when 334 judging the obtained results.” Evaluating rather than judging?
  • “we could consider the data of 2y-OS accurate.” -- > as accurate
  • - “data from investigated trials are missing and information on EBRT schedules.”: please revise

Author Response

We thanks the Reviewer for both the good comments and the observations. We tried to fulfill all the requirements, and hope that this improved the text.

An extensive English revision was also performed.

Following, we detail all the modifications performed following the Reviewer’s suggestions.

Abstract:  

  • treatments with different 223Ra injection activity levels à The word “different” is connected to “activity levels” and this is not syntactically correct: please revise

We modified the sentence as following: “Previously published studies combined external beam radiotherapy (EBRT) treatments with different activities of 223Ra”. We hope that this change may reflect the proper meaning.

  • do not abbreviate fraction in the abstract

We removed this abbreviation

 Page 2:

  • In 2013,

We modified the text a suggested

  • about five, compared

We modified the text a suggested

  • Relevant literature has been screened to obtain treatment outcomes in terms: please revise this sentence (“screened” and “obtain treatment outcomes” can be significantly improved)

We modified the sentence as following: “Relevant literature has been revised in order to extract information on  treatment outcomes in terms of both efficacy and safety from randomized phase II-III studies.” We hope that this change may reflect the proper meaning.

Page 3:

  • Please specify what MIRD formalism is (reference?)

We added the references

  • “was fitted with an exponential curve fitting of the spherical-like tumor model.” Please revise, without using tautologies.

We modified the sentence as following: “For each patient, the absorbed dose for each single lesion mass reported in Table 1 of ref. (14) was fitted with an exponential curve fitting. We used the spherical model for tumor dose evaluation.”

  • OLINDA/EXM software (version 1.1) (please add a reference).

We added the reference

Page 4:

  • 1: an approximation is made at the level of 215-Po: the sum of beta- and alfa decays is >100%. Please revise.

We thanks the reviewer for the observation. In fact, as the alpha decay of Po-215 is > 99.9% (99.977%) and the beta decay is < 0.001% (0.00023%), only the alpha decay of Po-215 was considered in our calculations. We added the following sentence: “As the branching ratio of beta 215-Po decay is <0.001%, this decay was omitted in the calculation and a full alpha decay in 211-Pb was assumed for dose estimation.”

  • Please consider “as follows” instead of “as following” here and along the manuscript

We modified it as suggested in the whole text.

  • Please specify what you refer to with “above parameters”

We refer to the parameters reported in Table 1. We modified the sentence as following: “Based on equation 3 and the parameters reported in Table 1, the …”

Page 5:

  • Please improve the sentence “explore the effect of dose on outcomes”

We modified the sentence as following: “Mixed-effect logistic-regression were used to model the effect of dose on outcomes”

Page 6:

  • Be aware that in studies.. please improve the sentence

We modified the sentence as following: “It should be noted that the 2y-OS information was extracted from each study with different modalities. Studies A and B reported the number of events or patient survived, thus 2y-OS was estimated assuming no censored patients. On the other hand, in studies C and D 2y-OS was extracted directly from Kaplan-Meier curves. Therefore, for studies C and D, censored data are included in the provided information.”

  • “without knowing if censoring is present”. please improve the sentence

We modified the sentence as following: “The proportion of surviving patients up to 2 years (2y-OS) is extracted from each study. It should be noted that the 2y-OS information was extracted from each study with different modalities. Studies A and B reported the number of events or patient survived, thus 2y-OS was estimated assuming no censored patients. On the other hand, in studies C and D 2y-OS was extracted directly from Kaplan-Meier curves. Therefore, for studies C and D, censored data are included in the provided information.”

Page 8:

  • “The hypothesis indicates that it is possible” if I have correctly interpreted the meaning, this is not an hypothesis, rather a result.

We modified the sentence as following: “The data reported in literature indicates that it is possible to increase tumoral cell killing and survival with combined α and photon treatments”

Page 9:

  • Missing verb in “Recently reported, dose-effect relationship for tumor control probability suggesting a minimum cut-off of 20 Gy for the absorbed dose to guarantee a stable disease, partial or complete response (23).”

We modified the sentence as following. “Recently reported, dose-effect relationship for tumor control probability suggests a minimum cut-off of 20 Gy for the absorbed dose to guarantee a stable disease, partial or complete response (23).”

  • “The association of neutropenia to combined EBRT+RN treatment with the delivered dose”, please specify what you refer to with “delivered doses”; as it is, it is ambiguous (with the doses delivered as shown in tab… etc)

We rephrase as following: “The association of neutropenia to the mean absorbed dose delivered with combined EBRT+RN treatment is confirmed in our study, highlighting that neutropenia incidence slightly increase with the given absorbed dose”.

  • Data+ plural

We correctes the sentence as following: “Our data show…”

  • What do you mean with “limited data available for pour analysis ” ?

We rephrased the sentence as following: “However, considering the limited data available and the impossibility of differentiating the EBRT…”

  • “We showed that increasing the mean absorbed dose to the target will increase 2y-OS” please revise

We modified the sentence as following: “We showed that increasing the mean absorbed dose to the target will increase 2y-OS up to about 50%. The same increase in mean absorbed dose is associated to an acceptable toxicity profile, that increase only up to about 20%.”

  • “combined RN 310 and EBRT therapy” -- > therapies

We modified the text as suggested

Page 10:

  • “be considered when 334 judging the obtained results.” Evaluating rather than judging?

We modified as suggested. We tank for the suggestion, as I describe better the concept.

  • “we could consider the data of 2y-OS accurate.” -- > as accurate

We modified the text as suggested.

  • - “data from investigated trials are missing and information on EBRT schedules.”: please revise

We modified the sentence as following: “In addition, patient-specific data (e.g. EBRT schedules, clinical information, …) from each considered study are missing.”
